# A Theoretically Grounded Application of Dropout in Recurrent Neural Networks

**Yarin Gal**

**Zoubin Ghahramani**

University of Cambridge
`{yg279,zg201}@cam.ac.uk`

## Abstract

Recurrent neural networks (RNNs) stand at the forefront of many recent developments in deep learning. Yet a major difficulty with these models is their tendency to overfit, with dropout shown to fail when applied to recurrent layers. Recent results at the intersection of Bayesian modelling and deep learning offer a Bayesian interpretation of common deep learning techniques such as dropout. This grounding of dropout in approximate Bayesian inference suggests an extension of the theoretical results, offering insights into the use of dropout with RNN models. We apply this new variational inference based dropout technique in LSTM and GRU models, assessing it on language modelling and sentiment analysis tasks. The new approach outperforms existing techniques, and to the best of our knowledge improves on the single model state-of-the-art in language modelling with the Penn Treebank (73.4 test perplexity). This extends our arsenal of variational tools in deep learning.

## 1 Introduction

Recurrent neural networks (RNNs) are sequence-based models of key importance for natural language understanding, language generation, video processing, and many other tasks [1–3]. The model's input is a sequence of symbols, where at each time step a simple neural network (*RNN unit*) is applied to a single symbol, as well as to the network's output from the previous time step. RNNs are powerful models, showing superb performance on many tasks, but overfit quickly. Lack of regularisation in RNN models makes it difficult to handle small data, and to avoid overfitting researchers often use early stopping, or small and under-specified models [4].

Dropout is a popular regularisation technique with deep networks [5, 6] where network units are randomly masked during training (*dropped*). But the technique has never been applied successfully to RNNs. Empirical results have led many to believe that noise added to recurrent layers (connections between RNN units) will be amplified for long sequences, and drown the signal [4]. Consequently, existing research has concluded that the technique should be used with the inputs and outputs of the RNN alone [4, 7–10]. But this approach still leads to overfitting, as is shown in our experiments.

Recent results at the intersection of Bayesian research and deep learning offer interpretation of common deep learning techniques through Bayesian eyes [11–16]. This Bayesian view of deep learning allowed the introduction of new techniques into the field, such as methods to obtain principled uncertainty estimates from deep learning networks [14, 17]. Gal and Ghahramani [14] for example showed that dropout can be interpreted as a variational approximation to the posterior of a Bayesian neural network (NN). Their variational approximating distribution is a mixture of two Gaussians with small variances, with the mean of one Gaussian fixed at zero. This grounding of dropout in approximate Bayesian inference suggests that an extension of the theoretical results might offer insights into the use of the technique with RNN models.

Here we focus on common RNN models in the field (LSTM [18], GRU [19]) and interpret these as probabilistic models, i.e. as RNNs with network weights treated as random variables, and with suitably defined likelihood functions. We then perform approximate variational inference in these probabilistic Bayesian models (which we will refer to as *Variational RNNs*). Approximating the posterior distribution over the weights with a mixture of Gaussians (with one component fixed at zero and small variances) will lead to a tractable optimisation objective. Optimising this objective is identical to performing a new variant of dropout in the respective RNNs.

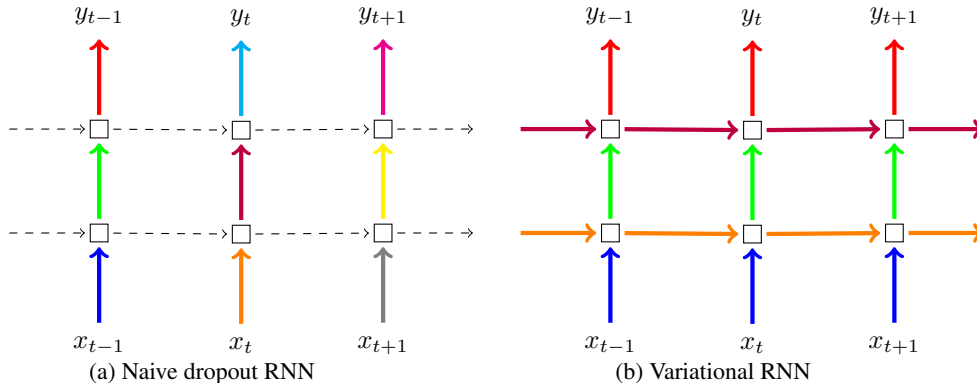

(a) Naive dropout RNN          (b) Variational RNN

Figure 1: **Depiction of the dropout technique following our Bayesian interpretation (right) compared to the standard technique in the field (left).** Each square represents an RNN unit, with horizontal arrows representing time dependence (recurrent connections). Vertical arrows represent the input and output to each RNN unit. Coloured connections represent dropped-out inputs, with different colours corresponding to different dropout masks. Dashed lines correspond to standard connections with no dropout. Current techniques (naive dropout, left) use different masks at different time steps, with no dropout on the recurrent layers. The proposed technique (Variational RNN, right) uses the same dropout mask at each time step, including the recurrent layers.

In the new dropout variant, we repeat the same dropout mask at each time step for both inputs, outputs, and recurrent layers (drop the same network units at each time step). This is in contrast to the existing ad hoc techniques where different dropout masks are sampled at each time step for the inputs and outputs alone (no dropout is used with the recurrent connections since the use of different masks with these connections leads to deteriorated performance). Our method and its relation to existing techniques is depicted in figure 1. When used with discrete inputs (i.e. words) we place a distribution over the word embeddings as well. Dropout in the word-based model corresponds then to randomly dropping word *types* in the sentence, and might be interpreted as forcing the model not to rely on single words for its task.

We next survey related literature and background material, and then formalise our approximate inference for the Variational RNN, resulting in the dropout variant proposed above. Experimental results are presented thereafter.

## 2   Related Research

In the past few years a considerable body of work has been collected demonstrating the negative effects of a naive application of dropout in RNNs' recurrent connections. Pachitariu and Sahani [7], working with language models, reason that noise added in the recurrent connections of an RNN leads to model instabilities. Instead, they add noise to the decoding part of the model alone. Bayer et al. [8] apply a deterministic approximation of dropout (fast dropout) in RNNs. They reason that with dropout, the RNN's dynamics change dramatically, and that dropout should be applied to the "non-dynamic" parts of the model – connections feeding from the hidden layer to the output layer. Pham et al. [9] assess dropout with handwriting recognition tasks. They conclude that dropout in recurrent layers disrupts the RNN's ability to model sequences, and that dropout should be applied to feed-forward connections and not to recurrent connections. The work by Zaremba, Sutskever, and Vinyals [4] was developed in parallel to Pham et al. [9]. Zaremba et al. [4] assess the performance of dropout in RNNs on a wide series of tasks. They show that applying dropout to the non-recurrent connections alone results in improved performance, and provide (as yet unbeaten) state-of-the-art results in language modelling on the Penn Treebank. They reason that without dropout only small models were used in the past in order to avoid overfitting, whereas with the application of dropout larger models can be used, leading to improved results. This work is considered a reference implementation by many (and we compare to this as a baseline below). Bluche et al. [10] extend on the previous body of work and perform exploratory analysis of the performance of dropout before, inside, and after the RNN's unit. They provide mixed results, not showing significant improvement on existing techniques. More recently, and done in parallel to this work, Moon et al. [20] suggested a new variant of dropout in RNNs in the speech recognition community. They randomly drop elements in the LSTM's internal cell $\mathbf{c}_t$ and use the same mask at every time step. This is the closest to our proposed approach (although fundamentally different to the approach we suggest, explained in §4.1), and we compare to this variant below as well.

Existing approaches are *based on an empirical experimentation* with different flavours of dropout, following a process of trial-and-error. These approaches have led many to believe that dropout cannot be extended to a large number of parameters within the recurrent layers, leaving them with no regularisation. In contrast to these conclusions, we show that it is possible to derive a variational inference based variant of dropout which successfully regularises such parameters, by grounding our approach in recent theoretical research.

## 3 Background

We review necessary background in Bayesian neural networks and approximate variational inference. Building on these ideas, in the next section we propose approximate inference in the probabilistic RNN which will lead to a new variant of dropout.

### 3.1 Bayesian Neural Networks

Given training inputs $\mathbf{X} = \{\mathbf{x}_1, \ldots, \mathbf{x}_N\}$ and their corresponding outputs $\mathbf{Y} = \{\mathbf{y}_1, \ldots, \mathbf{y}_N\}$, in Bayesian (parametric) regression we would like to infer parameters $\boldsymbol{\omega}$ of a function $\mathbf{y} = \mathbf{f}^{\boldsymbol{\omega}}(\mathbf{x})$ that are *likely to have generated* our outputs. What parameters are likely to have generated our data? Following the Bayesian approach we would put some *prior* distribution over the space of parameters, $p(\boldsymbol{\omega})$. This distribution represents our prior belief as to which parameters are likely to have generated our data. We further need to define a likelihood distribution $p(\mathbf{y}|\mathbf{x}, \boldsymbol{\omega})$. For classification tasks we may assume a softmax likelihood,

$$p\big(y = d|\mathbf{x}, \boldsymbol{\omega}\big) = \text{Categorical}\left(\exp(f_d^{\boldsymbol{\omega}}(\mathbf{x})) / \sum_{d'} \exp(f_{d'}^{\boldsymbol{\omega}}(\mathbf{x}))\right)$$

or a Gaussian likelihood for regression. Given a dataset $\mathbf{X}, \mathbf{Y}$, we then look for the *posterior* distribution over the space of parameters: $p(\boldsymbol{\omega}|\mathbf{X}, \mathbf{Y})$. This distribution captures how likely various function parameters are given our observed data. With it we can predict an output for a new input point $\mathbf{x}^*$ by integrating

$$p(\mathbf{y}^*|\mathbf{x}^*, \mathbf{X}, \mathbf{Y}) = \int p(\mathbf{y}^*|\mathbf{x}^*, \boldsymbol{\omega}) p(\boldsymbol{\omega}|\mathbf{X}, \mathbf{Y}) \mathrm{d}\boldsymbol{\omega}. \tag{1}$$

One way to define a distribution over a parametric set of functions is to place a prior distribution over a *neural network's* weights, resulting in a *Bayesian NN* [21, 22]. Given weight matrices $\mathbf{W}_i$ and bias vectors $\mathbf{b}_i$ for layer $i$, we often place standard matrix Gaussian prior distributions over the weight matrices, $p(\mathbf{W}_i) = \mathcal{N}(\mathbf{0}, \mathbf{I})$ and often assume a point estimate for the bias vectors for simplicity.

### 3.2 Approximate Variational Inference in Bayesian Neural Networks

We are interested in finding the distribution of weight matrices (parametrising our functions) that have generated our data. This is the posterior over the weights given our observables $\mathbf{X}, \mathbf{Y}$: $p(\boldsymbol{\omega}|\mathbf{X}, \mathbf{Y})$. This posterior is not tractable in general, and we may use variational inference to approximate it (as was done in [23–25, 12]). We need to define an approximating variational distribution $q(\boldsymbol{\omega})$, and then minimise the KL divergence between the approximating distribution and the full posterior:

$$\text{KL}\big(q(\boldsymbol{\omega})||p(\boldsymbol{\omega}|\mathbf{X}, \mathbf{Y})\big) \propto - \int q(\boldsymbol{\omega}) \log p(\mathbf{Y}|\mathbf{X}, \boldsymbol{\omega}) \mathrm{d}\boldsymbol{\omega} + \text{KL}(q(\boldsymbol{\omega})||p(\boldsymbol{\omega}))$$

$$= - \sum_{i=1}^{N} \int q(\boldsymbol{\omega}) \log p(\mathbf{y}_i|\mathbf{f}^{\boldsymbol{\omega}}(\mathbf{x}_i)) \mathrm{d}\boldsymbol{\omega} + \text{KL}(q(\boldsymbol{\omega})||p(\boldsymbol{\omega})). \tag{2}$$

We next extend this approximate variational inference to probabilistic RNNs, and use a $q(\boldsymbol{\omega})$ distribution that will give rise to a new variant of dropout in RNNs.

## 4 Variational Inference in Recurrent Neural Networks

In this section we will concentrate on simple RNN models for brevity of notation. Derivations for LSTM and GRU follow similarly. Given input sequence $\mathbf{x} = [\mathbf{x}_1, ..., \mathbf{x}_T]$ of length $T$, a simple RNN is formed by a repeated application of a function $\mathbf{f_h}$. This generates a hidden state $\mathbf{h}_t$ for time step $t$:

$$\mathbf{h}_t = \mathbf{f_h}(\mathbf{x}_t, \mathbf{h}_{t-1}) = \sigma(\mathbf{x}_t \mathbf{W_h} + \mathbf{h}_{t-1} \mathbf{U_h} + \mathbf{b_h})$$

for some non-linearity $\sigma$. The model output can be defined, for example, as $\mathbf{f_y}(\mathbf{h}_T) = \mathbf{h}_T \mathbf{W_y} + \mathbf{b_y}$. We view this RNN as a probabilistic model by regarding $\boldsymbol{\omega} = \{\mathbf{W_h}, \mathbf{U_h}, \mathbf{b_h}, \mathbf{W_y}, \mathbf{b_y}\}$ as random

variables (following normal prior distributions). To make the dependence on $\boldsymbol{\omega}$ clear, we write $\mathbf{f_y^\omega}$ for $\mathbf{f_y}$ and similarly for $\mathbf{f_h^\omega}$. We define our probabilistic model's likelihood as above (section 3.1). The posterior over random variables $\boldsymbol{\omega}$ is rather complex, and we use variational inference with approximating distribution $q(\boldsymbol{\omega})$ to approximate it.

Evaluating each sum term in eq. (2) above with our RNN model we get

$$\int q(\boldsymbol{\omega}) \log p(\mathbf{y}|\mathbf{f_y^\omega}(\mathbf{h}_T)) \mathrm{d}\boldsymbol{\omega} = \int q(\boldsymbol{\omega}) \log p\bigg(\mathbf{y}\bigg|\mathbf{f_y^\omega}\big(\mathbf{f_h^\omega}(\mathbf{x}_T, \mathbf{h}_{T-1})\big)\bigg) \mathrm{d}\boldsymbol{\omega}$$

$$= \int q(\boldsymbol{\omega}) \log p\bigg(\mathbf{y}\bigg|\mathbf{f_y^\omega}\big(\mathbf{f_h^\omega}(\mathbf{x}_T, \mathbf{f_h^\omega}(...\mathbf{f_h^\omega}(\mathbf{x}_1, \mathbf{h}_0)...))\big)\bigg) \mathrm{d}\boldsymbol{\omega}$$

with $\mathbf{h}_0 = \mathbf{0}$. We approximate this with Monte Carlo (MC) integration with a single sample:

$$\approx \log p\bigg(\mathbf{y}\bigg|\mathbf{f_y^{\widehat{\omega}}}\big(\mathbf{f_h^{\widehat{\omega}}}(\mathbf{x}_T, \mathbf{f_h^{\widehat{\omega}}}(...\mathbf{f_h^{\widehat{\omega}}}(\mathbf{x}_1, \mathbf{h}_0)...))\big)\bigg), \qquad \widehat{\boldsymbol{\omega}} \sim q(\boldsymbol{\omega})$$

resulting in an unbiased estimator to each sum term.

This estimator is plugged into equation (2) to obtain our minimisation objective

$$\mathcal{L} \approx -\sum_{i=1}^{N} \log p\bigg(\mathbf{y}_i\bigg|\mathbf{f_y^{\widehat{\omega}_i}}\big(\mathbf{f_h^{\widehat{\omega}_i}}(\mathbf{x}_{i,T}, \mathbf{f_h^{\widehat{\omega}_i}}(...\mathbf{f_h^{\widehat{\omega}_i}}(\mathbf{x}_{i,1}, \mathbf{h}_0)...))\big)\bigg) + \mathrm{KL}(q(\boldsymbol{\omega})\|p(\boldsymbol{\omega})). \qquad (3)$$

Note that for each sequence $\mathbf{x}_i$ we sample a new realisation $\widehat{\boldsymbol{\omega}}_i = \{\widehat{\mathbf{W}}_\mathbf{h}^i, \widehat{\mathbf{U}}_\mathbf{h}^i, \widehat{\mathbf{b}}_\mathbf{h}^i, \widehat{\mathbf{W}}_\mathbf{y}^i, \widehat{\mathbf{b}}_\mathbf{y}^i\}$, and that each symbol in the sequence $\mathbf{x}_i = [\mathbf{x}_{i,1}, ..., \mathbf{x}_{i,T}]$ is passed through the function $\mathbf{f_h^{\widehat{\omega}_i}}$ with *the same weight realisations* $\widehat{\mathbf{W}}_\mathbf{h}^i, \widehat{\mathbf{U}}_\mathbf{h}^i, \widehat{\mathbf{b}}_\mathbf{h}^i$ *used at **every time step*** $t \leq T$.

Following [17] we define our approximating distribution to factorise over the weight matrices and their rows in $\boldsymbol{\omega}$. For every weight matrix row $\mathbf{w}_k$ the approximating distribution is:

$$q(\mathbf{w}_k) = p\mathcal{N}(\mathbf{w}_k; \mathbf{0}, \sigma^2 I) + (1-p)\mathcal{N}(\mathbf{w}_k; \mathbf{m}_k, \sigma^2 I)$$

with $\mathbf{m}_k$ variational parameter (row vector), $p$ given in advance (the dropout probability), and small $\sigma^2$. We optimise over $\mathbf{m}_k$ the variational parameters of the random weight matrices; these correspond to the RNN's weight matrices in the standard view[1]. The KL in eq. (3) can be approximated as $L_2$ regularisation over the variational parameters $\mathbf{m}_k$ [17].

Evaluating the model output $\mathbf{f_y^{\widehat{\omega}}}(\cdot)$ with sample $\widehat{\boldsymbol{\omega}} \sim q(\boldsymbol{\omega})$ corresponds to randomly zeroing (masking) rows in each weight matrix $\mathbf{W}$ during the forward pass – i.e. performing dropout. Our objective $\mathcal{L}$ is identical to that of the standard RNN. In our RNN setting with a sequence input, each weight matrix row is randomly masked once, and importantly the same mask is used through all time steps.[2]

Predictions can be approximated by either propagating the mean of each layer to the next (referred to as the *standard dropout approximation*), or by approximating the posterior in eq. (1) with $q(\boldsymbol{\omega})$,

$$p(\mathbf{y}^*|\mathbf{x}^*, \mathbf{X}, \mathbf{Y}) \approx \int p(\mathbf{y}^*|\mathbf{x}^*, \boldsymbol{\omega})q(\boldsymbol{\omega})\mathrm{d}\boldsymbol{\omega} \approx \frac{1}{K}\sum_{k=1}^{K} p(\mathbf{y}^*|\mathbf{x}^*, \widehat{\boldsymbol{\omega}}_k) \qquad (4)$$

with $\widehat{\boldsymbol{\omega}}_k \sim q(\boldsymbol{\omega})$, i.e. by performing dropout at test time and averaging results (*MC dropout*).

## 4.1 Implementation and Relation to Dropout in RNNs

Implementing our approximate inference is identical to implementing dropout in RNNs with the *same network units dropped at each time step*, randomly dropping inputs, outputs, and recurrent connections. This is in contrast to existing techniques, where different network units would be dropped at different time steps, and no dropout would be applied to the recurrent connections (fig. 1).

Certain RNN models such as LSTMs and GRUs use different *gates* within the RNN units. For example, an LSTM is defined using four gates: "input", "forget", "output", and "input modulation",

$$\underline{\mathbf{i}} = \mathrm{sigm}\big(\mathbf{h}_{t-1}\mathbf{U}_i + \mathbf{x}_t\mathbf{W}_i\big) \qquad\qquad \underline{\mathbf{f}} = \mathrm{sigm}\big(\mathbf{h}_{t-1}\mathbf{U}_f + \mathbf{x}_t\mathbf{W}_f\big)$$

$$\underline{\mathbf{o}} = \mathrm{sigm}\big(\mathbf{h}_{t-1}\mathbf{U}_o + \mathbf{x}_t\mathbf{W}_o\big) \qquad\qquad \underline{\mathbf{g}} = \tanh\big(\mathbf{h}_{t-1}\mathbf{U}_g + \mathbf{x}_t\mathbf{W}_g\big)$$

$$\mathbf{c}_t = \underline{\mathbf{f}} \circ \mathbf{c}_{t-1} + \underline{\mathbf{i}} \circ \underline{\mathbf{g}} \qquad\qquad \mathbf{h}_t = \underline{\mathbf{o}} \circ \tanh(\mathbf{c}_t) \qquad\qquad (5)$$

with $\boldsymbol{\omega} = \{\mathbf{W}_i, \mathbf{U}_i, \mathbf{W}_f, \mathbf{U}_f, \mathbf{W}_o, \mathbf{U}_o, \mathbf{W}_g, \mathbf{U}_g\}$ weight matrices and $\circ$ the element-wise product. Here an internal state $\mathbf{c}_t$ (also referred to as *cell*) is updated additively.

Alternatively, the model could be re-parametrised as in [26]:

$$\begin{pmatrix} \underline{\mathbf{i}} \\ \underline{\mathbf{f}} \\ \underline{\mathbf{o}} \\ \underline{\mathbf{g}} \end{pmatrix} = \begin{pmatrix} \text{sigm} \\ \text{sigm} \\ \text{sigm} \\ \tanh \end{pmatrix} \left( \begin{pmatrix} \mathbf{x}_t \\ \mathbf{h}_{t-1} \end{pmatrix} \cdot \mathbf{W} \right) \qquad\qquad (6)$$

with $\boldsymbol{\omega} = \{\mathbf{W}\}$, $\mathbf{W}$ a matrix of dimensions $2K$ by $4K$ ($K$ being the dimensionality of $\mathbf{x}_t$). We name this parametrisation a *tied-weights* LSTM (compared to the *untied-weights* LSTM in eq. (5)).

Even though these two parametrisations result in the same *deterministic* model, they lead to different approximating distributions $q(\boldsymbol{\omega})$. With the first parametrisation one could use different dropout masks for different gates (even when the same input $\mathbf{x}_t$ is used). This is because the approximating distribution is placed over the matrices rather than the inputs: we might drop certain rows in one weight matrix $\mathbf{W}$ applied to $\mathbf{x}_t$ and different rows in another matrix $\mathbf{W}'$ applied to $\mathbf{x}_t$. With the second parametrisations we would place a distribution over the single matrix $\mathbf{W}$. This leads to a faster forward-pass, but with slightly diminished results as we will see in the experiments section.

In more concrete terms, we may write our dropout variant with the second parametrisation (eq. (6)) as

$$\begin{pmatrix} \underline{\mathbf{i}} \\ \underline{\mathbf{f}} \\ \underline{\mathbf{o}} \\ \underline{\mathbf{g}} \end{pmatrix} = \begin{pmatrix} \text{sigm} \\ \text{sigm} \\ \text{sigm} \\ \tanh \end{pmatrix} \left( \begin{pmatrix} \mathbf{x}_t \circ \mathbf{z}_{\mathbf{x}} \\ \mathbf{h}_{t-1} \circ \mathbf{z}_{\mathbf{h}} \end{pmatrix} \cdot \mathbf{W} \right) \qquad\qquad (7)$$

with $\mathbf{z}_{\mathbf{x}}, \mathbf{z}_{\mathbf{h}}$ random masks repeated at all time steps (and similarly for the parametrisation in eq. (5)).

In comparison, Zaremba et al. [4]'s variant replaces $\mathbf{z}_{\mathbf{x}}$ in eq. (7) with a time-dependent mask: $\mathbf{x}_t \circ \mathbf{z}_{\mathbf{x}}^t$ where $\mathbf{z}_{\mathbf{x}}^t$ is sampled anew every time step (whereas $\mathbf{z}_{\mathbf{h}}$ is removed and the recurrent connection $\mathbf{h}_{t-1}$ is not dropped). On the other hand, Moon et al. [20]'s variant changes eq. (5) by adapting the internal cell $\mathbf{c}_t = \mathbf{c}_t \circ \mathbf{z}_{\mathbf{c}}$ with the same mask $\mathbf{z}_{\mathbf{c}}$ used at all time steps. Note that unlike [20], by viewing dropout as an operation over the weights our technique trivially extends to RNNs and GRUs.

### 4.2   Word Embeddings Dropout

In datasets with continuous inputs we often apply dropout to the input layer – i.e. to the input vector itself. This is equivalent to placing a distribution over the weight matrix which follows the input and approximately integrating over it (the matrix is optimised, therefore prone to overfitting otherwise).

But for models with discrete inputs such as words (where every word is mapped to a continuous vector – a *word embedding*) this is seldom done. With word embeddings the input can be seen as either the word embedding itself, or, more conveniently, as a "one-hot" encoding (a vector of zeros with 1 at a single position). The product of the one-hot encoded vector with an embedding matrix $\mathbf{W}_E \in \mathbb{R}^{V \times D}$ (where $D$ is the embedding dimensionality and $V$ is the number of words in the vocabulary) then gives a word embedding. Curiously, this parameter layer is the largest layer in most language applications, yet it is often not regularised. Since the embedding matrix is optimised it can lead to overfitting, and it is therefore desirable to apply dropout to the one-hot encoded vectors. This in effect is identical to *dropping words at random* throughout the input sentence, and can also be interpreted as encouraging the model to not "depend" on single words for its output.

Note that as before, we randomly set rows of the matrix $\mathbf{W}_E \in \mathbb{R}^{V \times D}$ to zero. Since we repeat the same mask at each time step, we drop the same words throughout the sequence – i.e. we drop word types at random rather than word tokens (as an example, the sentence "the dog and the cat" might become "— dog and — cat" or "the — and the cat", but never "— dog and the cat"). A possible inefficiency implementing this is the requirement to sample $V$ Bernoulli random variables, where $V$ might be large. This can be solved by the observation that for sequences of length $T$, at most $T$ embeddings could be dropped (other dropped embeddings have no effect on the model output). For $T \ll V$ it is therefore more efficient to first map the words to the word embeddings, and only then to zero-out word embeddings based on their word type.

## 5   Experimental Evaluation

We start by implementing our proposed dropout variant into the Torch implementation of Zaremba et al. [4], that has become a reference implementation for many in the field. Zaremba et al. [4] have

set a benchmark on the Penn Treebank that to the best of our knowledge hasn't been beaten for the past 2 years. We improve on [4]'s results, and show that our dropout variant improves model performance compared to early-stopping and compared to using under-specified models. We continue to evaluate our proposed dropout variant with both LSTM and GRU models on a sentiment analysis task where labelled data is scarce. We finish by giving an in-depth analysis of the properties of the proposed method, with code and many experiments deferred to the appendix due to space constraints.

## 5.1 Language Modelling

We replicate the language modelling experiment of Zaremba, Sutskever, and Vinyals [4]. The experiment uses the Penn Treebank, a standard benchmark in the field. This dataset is considered a small one in the language processing community, with $887,521$ tokens (words) in total, making overfitting a considerable concern. Throughout the experiments we refer to LSTMs with the dropout technique proposed following our Bayesian interpretation as *Variational LSTMs*, and refer to existing dropout techniques as *naive dropout LSTMs* (different masks at different steps, applied to the input and output of the LSTM alone). We refer to LSTMs with no dropout as *standard LSTMs*.

We implemented a Variational LSTM for both the medium model of [4] (2 layers with 650 units in each layer) as well as their large model (2 layers with 1500 units in each layer). The only changes we've made to [4]'s setting are 1) using our proposed dropout variant instead of naive dropout, and 2) tuning weight decay (which was chosen to be zero in [4]). All other hyper-parameters are kept identical to [4]: learning rate decay was not tuned for our setting and is used following [4]. Dropout parameters were optimised with grid search (tying the dropout probability over the embeddings together with the one over the recurrent layers, and tying the dropout probability for the inputs and outputs together as well). These are chosen to minimise validation perplexity[3]. We further compared to Moon et al. [20] who only drop elements in the LSTM internal state using the same mask at all time steps (in addition to performing dropout on the inputs and outputs). We implemented their dropout variant with each model size, and repeated the procedure above to find optimal dropout probabilities (0.3 with the medium model, and 0.5 with the large model). We had to use early stopping for the large model with [20]'s variant as the model starts overfitting after 16 epochs. Moon et al. [20] proposed their dropout variant within the speech recognition community, where they did not have to consider embeddings overfitting (which, as we will see below, affect the recurrent layers considerably). We therefore performed an additional experiment using [20]'s variant together with our embedding dropout (referred to as *Moon et al. [20]+emb dropout*).

Our results are given in table 1. For the variational LSTM we give results using both the tied weights model (eq. (6)–(7), *Variational (tied weights)*), and without weight tying (eq. (5), *Variational (untied weights)*). For each model we report performance using both the standard dropout approximation (averaging the weights at test time – propagating the mean of each approximating distribution as input to the next layer), and using MC dropout (obtained by performing dropout at test time 1000 times, and averaging the model outputs following eq. (4), denoted *MC*). For each model we report average perplexity and standard deviation (each experiment was repeated 3 times with different random seeds and the results were averaged). Model training time is given in *words per second* (WPS).

It is interesting that using the dropout approximation, weight tying results in lower validation error and test error than the untied weights model. But with MC dropout the untied weights model performs much better. Validation perplexity for the large model is improved from [4]'s 82.2 down to 77.3 (with weight tying), or 77.9 without weight tying. Test perplexity is reduced from 78.4 down to 73.4 (with MC dropout and untied weights). To the best of our knowledge, these are currently the best single model perplexities on the Penn Treebank.

It seems that Moon et al. [20] underperform even compared to [4]. With no embedding dropout the large model overfits and early stopping is required (with no early stopping the model's validation perplexity goes up to 131 within 30 epochs). Adding our embedding dropout, the model performs much better, but still underperforms compared to applying dropout on the inputs and outputs alone.

Comparing our results to the non-regularised LSTM (evaluated with early stopping, giving similar performance as the early stopping experiment in [4]) we see that for either model size an improvement can be obtained by using our dropout variant. Comparing the medium sized Variational model to the large one we see that a significant reduction in perplexity can be achieved by using a larger model. This cannot be done with the non-regularised LSTM, where a larger model leads to worse results.

|  | Medium LSTM | | | Large LSTM | | |
|---|---|---|---|---|---|---|
|  | Validation | Test | WPS | Validation | Test | WPS |
| Non-regularized (early stopping) | 121.1 | 121.7 | 5.5K | 128.3 | 127.4 | 2.5K |
| Moon et al. [20] | 100.7 | 97.0 | 4.8K | 122.9 | 118.7 | 3K |
| Moon et al. [20] +emb dropout | 88.9 | 86.5 | 4.8K | 88.8 | 86.0 | 3K |
| Zaremba et al. [4] | 86.2 | 82.7 | 5.5K | 82.2 | 78.4 | 2.5K |
| Variational (tied weights) | $81.8 \pm 0.2$ | $79.7 \pm 0.1$ | 4.7K | $77.3 \pm 0.2$ | $75.0 \pm 0.1$ | 2.4K |
| Variational (tied weights, MC) | – | $79.0 \pm 0.1$ | – | – | $74.1 \pm 0.0$ | – |
| Variational (untied weights) | $81.9 \pm 0.2$ | $79.7 \pm 0.1$ | 2.7K | $77.9 \pm 0.3$ | $75.2 \pm 0.2$ | 1.6K |
| Variational (untied weights, MC) | – | $\mathbf{78.6 \pm 0.1}$ | – | – | $\mathbf{73.4 \pm 0.0}$ | – |

Table 1: Single model perplexity (on test and validation sets) for the Penn Treebank language modelling task. Two model sizes are compared (a medium and a large LSTM, following [4]'s setup), with number of processed words per second (WPS) reported. Both dropout approximation and MC dropout are given for the test set with the Variational model. A common approach for regularisation is to reduce model complexity (necessary with the non-regularised LSTM). With the Variational models however, a significant reduction in perplexity is achieved by using larger models.

This shows that reducing the complexity of the model, a possible approach to avoid overfitting, actually leads to a worse fit when using dropout.

We also see that the tied weights model achieves very close performance to that of the untied weights one when using the dropout approximation. Assessing model run time though (on a Titan X GPU), we see that tying the weights results in a more time-efficient implementation. This is because the single matrix product is implemented as a single GPU kernel, instead of the four smaller matrix products used in the untied weights model (where four GPU kernels are called sequentially). Note though that a low level implementation should give similar run times.

We further experimented with a model averaging experiment following [4]'s setting, where several large models are trained independently with their outputs averaged. We used Variational LSTMs with MC dropout following the setup above. Using 10 Variational LSTMs we improve [4]'s test set perplexity from 69.5 to 68.7 – obtaining identical perplexity to [4]'s experiment with 38 models.

Lastly, we report validation perplexity with reduced learning rate decay (with the medium model). Learning rate decay is often used for regularisation by setting the optimiser to make smaller steps when the model starts overfitting (as done in [4]). By removing it we can assess the regularisation effects of dropout alone. As can be seen in fig. 2, even with early stopping, Variational LSTM achieves lower perplexity than naive dropout LSTM and standard LSTM. Note though that a significantly lower perplexity for all models can be achieved with learning rate decay scheduling as seen in table 1

## 5.2 Sentiment Analysis

We next evaluate our dropout variant with both LSTM and GRU models on a sentiment analysis task, where labelled data is scarce. We use MC dropout (which we compare to the dropout approximation further in appendix B), and untied weights model parametrisations.

We use the raw Cornell film reviews corpus collected by Pang and Lee [27]. The dataset is composed of 5000 film reviews. We extract consecutive segments of $T$ words from each review for $T = 200$, and use the corresponding film score as the observed output $y$. The model is built from one embedding layer (of dimensionality 128), one LSTM layer (with 128 network units for each gate; GRU setting is built similarly), and finally a fully connected layer applied to the last output of the LSTM (resulting in a scalar output). We use the Adam optimiser [28] throughout the experiments, with batch size 128, and MC dropout at test time with 10 samples.

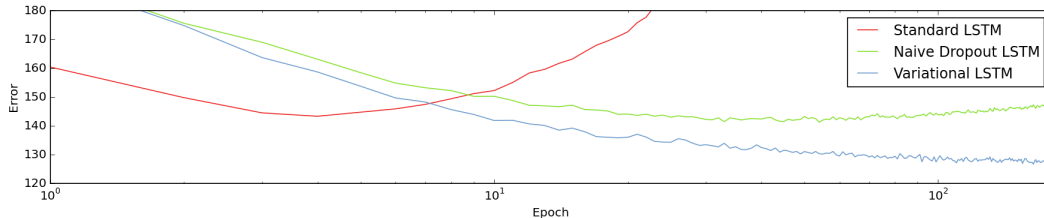

Figure 2: Medium model validation perplexity for the Penn Treebank language modelling task. Learning rate decay was reduced to assess model overfitting using dropout alone. Even with early stopping, Variational LSTM achieves lower perplexity than naive dropout LSTM and standard LSTM. Lower perplexity for all models can be achieved with learning rate decay scheduling, seen in table 1.

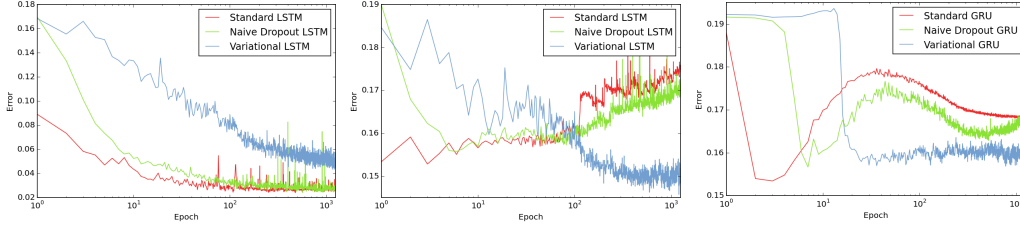

(a) LSTM train error: variational, naive dropout, and standard LSTM.

(b) LSTM test error: variational, naive dropout, and standard LSTM.

(c) GRU test error: variational, naive dropout, and standard LSTM.

Figure 3: Sentiment analysis error for *Variational LSTM / GRU* compared to *naive dropout LSTM / GRU* and *standard LSTM / GRU* (with no dropout).

The main results can be seen in fig. 3. We compared Variational LSTM (with our dropout variant applied with each weight layer) to standard techniques in the field. Training error is shown in fig. 3a and test error is shown in fig. 3b. Optimal dropout probabilities and weight decay were used for each model (see appendix B). It seems that the only model not to overfit is the Variational LSTM, which achieves lowest test error as well. Variational GRU test error is shown in fig. 14 (with loss plot given in appendix B). Optimal dropout probabilities and weight decay were used again for each model. Variational GRU avoids overfitting to the data and converges to the lowest test error. Early stopping in this dataset will result in smaller test error though (lowest test error is obtained by the non-regularised GRU model at the second epoch). It is interesting to note that standard techniques exhibit peculiar behaviour where test error repeatedly decreases and increases. This behaviour is not observed with the Variational GRU. Convergence plots of the loss for each model are given in appendix B.

We next explore the effects of dropping-out different parts of the model. We assessed our Variational LSTM with different combinations of dropout over the embeddings ($p_E = 0, 0.5$) and recurrent layers ($p_U = 0, 0.5$) on the sentiment analysis task. The convergence plots can be seen in figure 4a. It seems that without both strong embeddings regularisation and strong regularisation over the recurrent layers the model would overfit rather quickly. The behaviour when $p_U = 0.5$ and $p_E = 0$ is quite interesting: test error decreases and then increases before decreasing again. Also, it seems that when $p_U = 0$ and $p_E = 0.5$ the model becomes very erratic.

Lastly, we tested the performance of Variational LSTM with different recurrent layer dropout probabilities, fixing the embedding dropout probability at either $p_E = 0$ or $p_E = 0.5$ (figs. 4b-4c). These results are rather intriguing. In this experiment all models have converged, with the loss getting near zero (not shown). Yet it seems that with no embedding dropout, a higher dropout probability within the recurrent layers leads to overfitting! This presumably happens because of the large number of parameters in the embedding layer which is not regularised. Regularising the embedding layer with dropout probability $p_E = 0.5$ we see that a higher recurrent layer dropout probability indeed leads to increased *robustness* to overfitting, as expected. These results suggest that embedding dropout can be of crucial importance in some tasks.

In appendix B we assess the importance of weight decay with our dropout variant. Common practice is to remove weight decay with naive dropout. Our results suggest that weight decay plays an important role with our variant (it corresponds to our prior belief of the distribution over the weights).

# 6 Conclusions

We presented a new technique for recurrent neural network regularisation. Our RNN dropout variant is theoretically motivated and its effectiveness was empirically demonstrated.

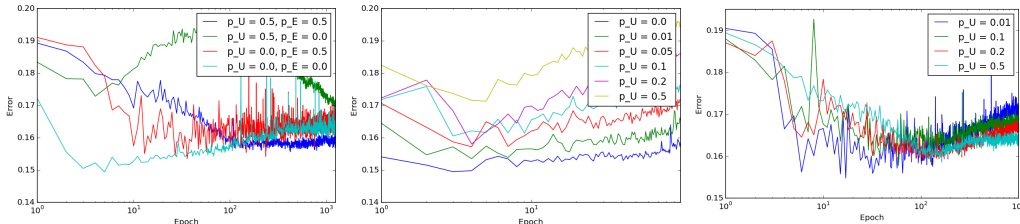

(a) Combinations of $p_E = 0, 0.5$ with $p_U = 0, 0.5$.

(b) $p_U = 0, ..., 0.5$ with fixed $p_E = 0$.

(c) $p_U = 0, ..., 0.5$ with fixed $p_E = 0.5$.

Figure 4: Test error for Variational LSTM with various settings on the sentiment analysis task. Different dropout probabilities are used with the recurrent layer ($p_U$) and embedding layer ($p_E$).

## Footnotes

[1]Graves et al. [26] further factorise the approximating distribution over the elements of each row, and use a Gaussian approximating distribution with each element (rather than a mixture); the approximating distribution above seems to give better performance, and has a close relation with dropout [17].

[2]In appendix A we discuss the relation of our dropout interpretation to the ensembling one.

[3]Optimal probabilities are 0.3 and 0.5 respectively for the large model, compared [4]'s 0.6 dropout probability, and 0.2 and 0.35 respectively for the medium model, compared [4]'s 0.5 dropout probability.

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
