[Supplementary Material]

## A Bayesian versus ensembling interpretation of dropout

Apart from our Bayesian approximation interpretation, dropout in *deep* networks can also be seen as following an ensembling interpretation [6]. This interpretation also leads to MC dropout at test time. But the ensembling interpretation does not determine whether the ensemble should be over the network units or the weights. For example, in an RNN this view will *not* lead to our dropout variant, unless the ensemble is *defined to tie the weights of the network* ad hoc. This is in comparison to the Bayesian approximation view where the weight tying is forced by the probabilistic interpretation of the model.

## B Sentiment analysis – further experiments

Sentiment analysis hyper-parameters were obtained by evaluating each model with dropout probabilities $0.25$ and $0.5$, and weight decays ranging from $10^{-6}$ to $10^{-4}$. The optimal setting for Variational LSTM is dropout probabilities $0.25$ and weight decay $10^{-3}$, and for naive dropout LSTM the dropout probabilities are $0.5$ (no weight decay is used in reference implementations of naive dropout LSTM [4]).

We assess the *dropout approximation* in Variational LSTMs. The dropout approximation is often used in deep networks as means of approximating the MC estimate. In the approximation we replace each weight matrix $\mathbf{M}$ by $p\mathbf{M}$ where $p$ is the dropout probability, and perform a deterministic pass through the network (without dropping out units). This can be seen as propagating the mean of the random variables $\mathbf{W}$ through the network [17]. The approximation has been shown to work well for deep networks [6], yet it fails with convolution layers [14]. We assess the approximation empirically with our Variational LSTM model, repeating the first experiment with the approximation used at test time instead of MC dropout. The results can be seen in fig. 9. It seems that the approximation gives a good estimate to the test error, similar to the results in figure 4a.

We further tested the Variational LSTM model with different weight decays, observing the effects of different values for these. Note that weight decay is applied to all layers, including the embedding layer. In figure 6 we can see that higher weight decay values result in lower test error, with significant differences for different weight decays. This suggests that weight decay still plays an important role even when using dropout (whereas common practice is to remove weight decay with naive dropout). Note also that the weight decay can be optimised (together with the dropout parameters) as part of the variational approximation. This is not done in practice in the deep learning community, where grid-search or Bayesian optimisation are often used for these instead.

Testing the Variational LSTM with different sequence lengths (with sequences of lengths $T = 20, 50, 200, 400$) we can see that sequence length has a strong effect on model performance as well (fig. 7). Longer sequences result in much better performance but with the price of longer convergence time. We hypothesised that the diminished performance on shorter sequences is caused by the high dropout probability on the embeddings. But a follow-up experiment with sequence lengths 50 and 200, and different embedding dropout probabilities, shows that lower dropout probabilities result in even worse model performance (figures 8 and 5).

In fig. 10a we see how different dropout probabilities and weight decays affect GRU model performance.

Figure 5: $p_E = 0, ..., 0.5$ with fixed $p_U = 0.5$.

Figure 6: Test error for Variational LSTM with different weight decays.

Figure 7: Variational LSTM test error for different sequence lengths ($T = 20, 50, 200, 400$ cut-offs).

Figure 8: Test error for various embedding dropout probabilities, with sequence length 50.

Figure 9: Dropout approximation in Variational LSTM with different dropout probabilities.

(a) Various Variational GRU model configurations

Figure 10: **Sentiment analysis error for *Variational GRU* compared to *naive dropout GRU* and *standard GRU* (with no dropout).** Test error for the different models (left) and for different Variational GRU configurations (right).

We compare naive dropout LSTM to Variational LSTM with dropout probability in the recurrent layers set to zero: $p_U = 0$ (referred to as *dropout LSTM*). Both models apply dropout to the input and outputs of the LSTM alone, with no dropout applied to the embeddings. Naive dropout LSTM uses different masks at different time steps though, tied across the gates, whereas dropout LSTM uses the same mask at different time steps. The test error for both models can be seen in fig. 11. It seems that without dropout over the recurrent layers and embeddings both models overfit, and in fact result in identical performance.

Next, we assess the dropout approximation in the GRU model. The approximation seems to give similar results to MC dropout in the GRU model (fig. 12).

Figure 11: *Naive dropout LSTM* uses different dropout masks at each time step, whereas *Dropout LSTM* uses the same mask at each time step. Both models apply dropout to the inputs and outputs alone, and result in identical performance.

Figure 12: GRU dropout approximation

Lastly, we plot the train loss for various models from the main body of the paper. All models have converged, with a stable train loss.

Figure 13: Train loss (as a func-
tion of batches) for figure 3a

Figure 14: GRU train loss (as a
function of batches) (figure 14)

Figure 15: Train loss (as a func-
tion of batches) for figure 4a

## C   Code

An efficient Theano [29] implementation of the method above into Keras [30] is as simple as:

```
def get_output(self, train=False):
    X = self.get_input(train)

    retain_prob_W = 1. - self.p_W[0]
    retain_prob_U = 1. - self.p_U[0]
    B_W = self.srng.binomial((4, X.shape[1], self.input_dim),
        p=retain_prob_W, dtype=theano.config.floatX)
    B_U = self.srng.binomial((4, X.shape[1], self.output_dim),
        p=retain_prob_U, dtype=theano.config.floatX)

    xi = T.dot(X * B_W[0], self.W_i) + self.b_i
    xf = T.dot(X * B_W[1], self.W_f) + self.b_f
    xc = T.dot(X * B_W[2], self.W_c) + self.b_c
    xo = T.dot(X * B_W[3], self.W_o) + self.b_o

    [outputs, memories], updates = theano.scan(
        self._step,
        sequences=[xi, xf, xo, xc],
        outputs_info=[
            T.unbroadcast(alloc_zeros_matrix(X.shape[1], self.output_dim), 1),
            T.unbroadcast(alloc_zeros_matrix(X.shape[1], self.output_dim), 1)
        ],
        non_sequences=[self.U_i, self.U_f, self.U_o, self.U_c, B_U],
        truncate_gradient=self.truncate_gradient)

    return outputs[-1]

def _step(self,
        xi_t, xf_t, xo_t, xc_t,
        h_tm1, c_tm1,
        u_i, u_f, u_o, u_c, B_U):
    i_t = self.inner_activation(xi_t + T.dot(h_tm1 * B_U[0], u_i))
    f_t = self.inner_activation(xf_t + T.dot(h_tm1 * B_U[1], u_f))
    c_t = f_t * c_tm1 + i_t * self.activation(xc_t + T.dot(h_tm1 * B_U[2], u_c))
    o_t = self.inner_activation(xo_t + T.dot(h_tm1 * B_U[3], u_o))
    h_t = o_t * self.activation(c_t)
    return h_t, c_t
```