[Reviews · NeurIPS 2016]

Reviewer 1

Summary

A dropout recipe for RNNs, based on recent work (ICML 2016) by the author on Bayesian-motivated dropout, is presented. In this formulation weights are treated as random variables and dropout is formulated via inference of a bi-modal approximate posterior distribution on rows of weights, with one of the two Gaussians in the posterior having a fixed mean at zero. This implies that outputs with shared weights share the same dropout mask, and so implies that the dropout on recurrent weights, inputs, and outputs should share the same dropout mask for all t.

Qualitative Assessment

Results on Penn TreeBank, a small LM task, establish a new state-of-the-art in terms of perplexity. Results on a sentiment analysis task are less convincing, as even early-stopping outperforms all dropout methods. If the size of the model was increased, could further gains be realized with the dropout-based approaches? Another concern is that the differences between a related method published by Moon et al.[19] and this work on not explicit enough. Moon's approach apparently performs very badly on Penn Treebank---is this only because the input and output masks are also not tied? Some additional commentary on this result and any other subtleties that distinguish the methods would be appreciated. A major weakness of the paper, particularly considering the fact that the techniques described have already been published, is the lack of further experimental evidence of the efficacy of the approach. At this point there is certainly no shortage of more significant benchmarks whose state-of-the-art systems are RNNs. From a performance standpoint this could be a very influential paper, but based on the current experiments, the true potential of the approach is not clear. Other minor comments: - hypers -> hyperparameters - why only one dropout rate for Moon's approach, while Variational dropout gets an input-output and a recurrent dropout parameter?

Confidence in this Review

2-Confident (read it all; understood it all reasonably well)


Reviewer 2

Summary

This paper applies a recently-proposed variational interpretation of binary dropout, to recurrent neural networks. Optimization with this type of binary dropout and L2 regularization of the weights, is equivalent to optimization of the variational bound with a Gaussian prior p(w) over the weights and an approximate posterior q(w) over each column of the weight matrix that is a mixture of two Dirac delta's (Gaussians with infinitesimal variance), one at the origin 0 and one at a location specified by the values of the weight matrix. So in this interpretation, the weights in a neural net, in this case RNN, act as the variational parameters to the variational posterior from which we sample.

Qualitative Assessment

- The paper is clearly written and I could not find mistakes in the proofs are derivations and proofs. This paper applies this principle to LSTMs. Two "variational LSTMs" are proposed: one with a giant weight matrix with a single dropout mask ("tied weights"), and one with individual weight matrices with individual dropout masks ("untied weights"). Experimentally, the method works well. Tying or untying the weights doesn't make a large difference, but both improve upon the pre-existing methods. - Experiments should be expanded. The dropout rates in this work are hand-tuned, while we know that they can be optimized in principle: in case of Gaussian dropout, as described in "Variational Dropout and the local reparameterization trick", or in case of Bernoullis dropout, through biased or unbiased gradient ascent using reinforce and baselines. Why does this paper lack any efforts in this direction? - The work from Moon et al is very similar, but does not provide the same theoretical foundation. It is encouraging that the results are this much better.

Confidence in this Review

3-Expert (read the paper in detail, know the area, quite certain of my opinion)


Reviewer 3

Summary

The paper gives a Bayesian interpretation of dropout applying on recurrent neural network, as an extension of a previous work for the feedforward case.

Qualitative Assessment

The paper gives a Bayesian interpretation of a potential dropout-like technique for training recurrent neural nets. This research direction is good for its potential influences on theoretical aspect for understanding both RNN and dropout. However, under the proposed framework, the reason why this technique is useful is still a bit unclear. In particular, the paper proposes a mixture prior on the row of weights (line 129), without explaining the benefit of doing so, besides resulting in an interpretation of dropout. Also, much of the interpretation is a simple extension of the previously proposed interpretation of dropout in feedforward case (Gal and Ghahramani, ICML2016) and can hardly be considered as a significant novelty, and the empirical novelty is also diminished because of the previous paper from Moon et al. The experiment section is also a bit weak. Since the dropout on RNN is not a common technique as in feedfoward case, the authors should show that such method really works broadly. However, in the experiments the model is only compared with other RNN+dropout results, while we are not even convinced of doing dropout (in recurrent direction) for RNNs, more model variants should probably be checked here. Besides, it's good to see it achieves SOTA on PTB word-level, while PTB is a fairly small dataset. It would be much better if there are some results on much larger scale RNN problems.

Confidence in this Review

2-Confident (read it all; understood it all reasonably well)


Reviewer 4

Summary

This paper proposes a novel approach of dropout for Recurrent Neural Networks (RNN). RNN in this paper is formulated as a Bayesian Neural Network. The authors use Variational Inference to train the RNN. They use Gaussian distributions as variational distributions for weight parameters. A novel dropout can be applied to this approach by using 0 mean Gaussian distribution as a part of variational distributions. Previous empirical works suggested that dropout over recurrent layer was not effective. This paper suggests that bayesian approach is effective for dropout over recurrent layer. In addition, the authors apply the novel dropout to weight parameters for word embedding. Experimental results show that this approach is effective to avoid overfitting.

Qualitative Assessment

Line 205: Previous method has 0 for weight decay. However, authors tune the weight decay for their proposed method. Does the proposed method outperform the previous method when the weight decay has 0? Tuning weight decay may have an enormous influence on the experimental results. I think that experimental conditions should be same except for the part of the dropout that authors propose. Line 287-288: Why does test error decrease and then increase before decreasing again? Figure4(a): When P_U = 0.5 and P_E = 0.0, test error can decrease after the number of epochs is 10^3. The authors should show the test error after the number of epochs is 10^3. Figure 4(b): Why is the number of epochs less than other experiments such as Figure 4(a) and 4(c)? Figure 4(a) shows that test error drastically change after the number of epochs is 10^2. Figure 4(b) should show test error after the number of epochs is 10^2. Figure 4(b) suggests that dropout for recurrent layer gives high test error. I think that dropout for recurrent layer is not effective even if the proposed method is applied. In addition, Figure 4(a) and 4(b) suggest that dropout for embedding layer is more important than dropout for recurrent layer. Other: Authors should unify the description when they refer figures in this paper. For example, “fig” is used in Line 272 but “figure” is used in Line 285.

Confidence in this Review

2-Confident (read it all; understood it all reasonably well)


Reviewer 5

Summary

The paper proposes a novel way to regularize recurrent neural networks through Bayesian view of dropout. The authors show improved performance on Penn Treebank language modelling task over other single network models. They also demonstrate the proposed method on a sentiment analysis task.

Qualitative Assessment

While the theoretical foundations are sound, the experimental evaluation is not complete. In particular, the sentiment analysis is not convincing. In section 5.2, many qualitative conclusions are drawn purely from learning curves (Figures 3 and 4) which might not be statistically representative. For example, in Figure 3c, it is not clear which of the method is actually the best, in particular since the standard and naive implementation have peculiar learning curves. Moreover, in Figure 4, it is hard to draw any conclusions due to high variance. To make the analysis more competent, you could for example consider using cross validation over the entire data set and report the results as an average over the folds. Also please include the initial error in the graphs, or describe how the error should be interpreted. The experiments with the language modeling task are more convincing. It would be interested to see how your model would perform if it was trained in standard deterministic manner instead of minimizing (3) but using exactly the same dropout masks as in Variational RNN.

Confidence in this Review

2-Confident (read it all; understood it all reasonably well)